# ON THE IMPORTANCE OF ARCHITECTURES AND HYPER-PARAMETERS FOR FAIRNESS IN FACE RECOGNITION

## ABSTRACT

Face recognition systems are deployed across the world by government agencies and contractors for sensitive and impactful tasks, such as surveillance and database matching. Despite their widespread use, these systems are known to exhibit bias across a range of sociodemographic dimensions, such as gender and race. Nonetheless, an array of works proposing pre-processing, training, and post-processing methods have failed to close these gaps. Here, we take a very different approach to this problem, identifying that both architectures and hyperparameters of neural networks are instrumental in reducing bias. We first run a large-scale analysis of the impact of architectures and training hyperparameters on several common fairness metrics and show that the implicit convention of choosing high-accuracy architectures may be suboptimal for fairness. Motivated by our findings, we run the first neural architecture search for fairness, jointly with a search for hyperparameters. We output a suite of models which Pareto-dominate all other competitive architectures in terms of accuracy and fairness. Furthermore, we show that these models transfer well to other face recognition datasets with similar and distinct protected attributes. We release our code and raw result files so that researchers and practitioners can replace our fairness metrics with a bias measure of their choice.

## 1 INTRODUCTION

Face recognition is regularly deployed across the world by government agencies for tasks including surveillance, employment, and housing decisions. However, recent studies have shown that face recognition systems exhibit disparity in accuracy based on race and gender (Grother et al., 2019; Raji et al., 2020; Raji & Fried, 2021; Learned-Miller et al., 2020). For example, some face recognition models were 10 or 100 times more likely to give false positives for Black or Asian people, compared to white people (Allyn, 2020). This bias has already led to multiple false arrests and jail time for innocent Black men in the USA (Hill, 2020a).

Motivated by the discovery of bias in face recognition and other models deployed in real-world applications, dozens of definitions for fairness have been proposed (Verma & Rubin, 2018), and many pre-processing, training, and post-processing techniques have been developed to mitigate model bias. However, these techniques have fallen short of de-biasing face recognition systems, and training fair models in this setting demands addressing several technical challenges (Cherepanova et al., 2021b).

While existing methods for de-biasing face recognition systems use a fixed neural network architecture and training hyperparameter setting, we instead ask a fundamental question which has received little attention: *does model-bias stem from the architecture and hyperparameters?* We further ask whether we can exploit the extensive research in the fields of neural architecture search (NAS) (Elsken et al., 2019) and hyperparameter optimization (HPO) (Feurer & Hutter, 2019) to search for models that achieve a desired trade-off between model-bias and accuracy.

In this work, we take the first step towards answering these questions. To this end, we conduct the first large-scale analysis of the relationship between hyperparameters, architectures, and bias. We train a diverse set of 29 architectures, ranging from ResNets (He et al., 2016b) to vision transformers (Dosovitskiy et al., 2020; Liu et al., 2021) to Gluon Inception V3 (Szegedy et al., 2016) to MobileNetV3 (Howard et al., 2019) on CelebA (Liu et al., 2015), for a total of 88 493 GPU hours. We train each of these architectures across different head, optimizer, and learning rate combinations. Our results show that different architectures learn different inductive biases from the same dataset. We conclude

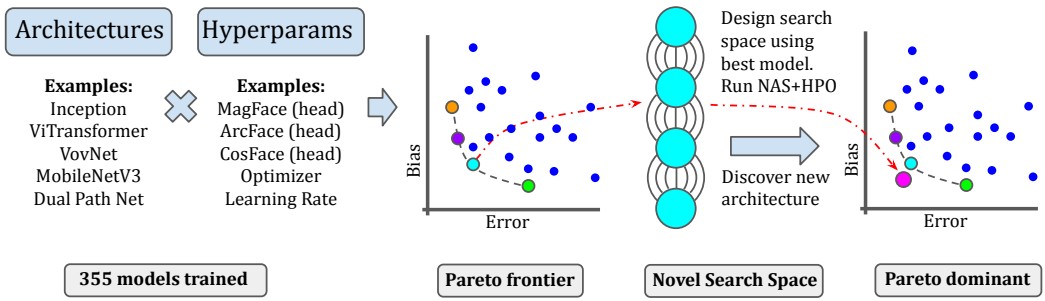

Figure 1: Overview of our methodology.

that the implicit convention of choosing the highest-accuracy architectures can be detrimental to fairness, and suggest that architecture and hyperparameters play a significant role in determining the fairness-accuracy tradeoff.

Next, we exploit this observation in order to design architectures with a better fairness-accuracy tradeoff. We initiate the study of NAS for fairness; specifically, we run NAS+HPO to jointly optimize fairness and accuracy. To tackle this problem, we construct a search space based on the highest-performing architecture from our analysis. We use the Sequential Model-based Algorithm Configuration method (SMAC (Lindauer et al., 2022)), for multi-objective architecture and hyperparameter search. We discover a Pareto frontier of face recognition models that outperform existing state-of-the-art models on both accuracy and multiple fairness metrics. An overview of our methodology can be found in Figure 1. We release all of our code and raw results at https://anonymous.4open.science/r/FR-NAS-92EC so that users can adapt our work to any bias measure of their choice.

**Our contributions** We summarize our main contributions below:

- We provide a new bias mitigation strategy which identifies that architectures have a profound influence on fairness, and then exploits that insight in order to design more fair architectures via Neural Architecture Search and Hyperparameter Optimization.
- We conduct a large-scale study of 29 architectures, each trained across a variety of hyperparameters, totalling 88 493 GPU hours, showing that architectures and hyperparameters have a big impact on fairness. We then conduct the first neural architecture search for fairness, jointly with hyperparameter optimization and optimizing for accuracy — culminating in a set of architectures which Pareto-dominate all models in a large set of modern architectures.
- Our new architectures outperform the current state of the art architecture, ArcFace (Deng et al., 2019), when training and testing CelebA and VGGFace2, and when training on CelebA and testing on other face recognition datasets (LFW, CFP-FP, CPLFW, AgeDB, and CALFW). Furthermore our architectures transfer well across different protected attributes Section 4.3.1.

## 2 BACKGROUND AND RELATED WORK

While our work is the first to leverage neural architecture search (NAS) to build fair models, a body of prior work exists in the fields of NAS and face recognition, and we discuss it here.

**Face Recognition.** Face recognition tasks fall into two categories: verification and identification. *Verification* asks whether the person in a source image is the same person as in the target image; this is a one-to-one comparison. *Identification* instead asks whether a given person in a source image appears within a gallery composed of many target identities and their associated images; this is a one-to-many comparison. Novel techniques in face recognition tasks, such as ArcFace (Wang et al., 2018), CosFace (Deng et al., 2019), and MagFace (Meng et al., 2021), use deep networks (often called the *backbone*) to extract feature representations of faces and then compare those to match individuals (with mechanisms called the *head*). Generally, *backbones* take the form of image feature extractors and *heads* resemble MLPs with specialized loss functions. Often, the term "head" refers to both the last layer of the network and the loss function. We focus our analysis on identification,

and we focus our evaluation on examining how close images of similar identities are in the feature space of trained models, since the technology relies on this feature representation to differentiate individuals. An overview of recent research on these topics can be found in Wang & Deng (2018).

**Sociodemographic Disparities in Face Recognition.** The existence of differential performance of face recognition on population groups and subgroups has been explored in a variety of settings. Earlier work (e.g., Klare et al., 2012; O'Toole et al., 2012) focuses on single-demographic effects (specifically, race and gender) in pre-deep-learning face detection and recognition. Buolamwini & Gebru (2018) uncover unequal performance at the phenotypic subgroup level in, specifically, a gender classification task powered by commercial systems. That work, typically referred to as "Gender Shades", continues to be hugely impactful both within academia and at the industry level. Indeed, Raji & Buolamwini (2019) provide a follow-up analysis – exploring the impact of the public disclosures in Buolamwini & Gebru (2018) – where they find that named companies (IBM, Microsoft, and Megvii) updated their APIs within a year to address some concerns that had surfaced. Further research continues to show that commercial face recognition systems still have sociodemogrpahic disparities in many complex and pernicious ways (Drozdowski et al., 2020; Dooley et al., 2021; Jaiswal et al., 2022; Dooley et al., 2022; Jaiswal et al., 2022).

In this work, we focus on *measuring* the sociodemographic disparities across neural architectures and hyperparameter settings, and finding the Pareto frontier of face recognition performance and bias for current and novel architectures. Our work searches for architectures and hyperparameters which improve the undesired disparities. Previous work focuses on "fixing" unfair systems and can be split into three (or arguably four (Savani et al., 2020)) focus areas: preprocessing (e.g., Feldman et al., 2015; Ryu et al., 2018; Quadrianto et al., 2019; Wang & Deng, 2020), inprocessing (e.g., Zafar et al., 2017; 2019; Donini et al., 2018; Goel et al., 2018; Padala & Gujar, 2020; Wang & Deng, 2020; Martinez et al., 2020; Nanda et al., 2021; Diana et al., 2020; Lahoti et al., 2020), and post-processing (e.g., Hardt et al., 2016; Wang et al., 2020b).

**Neural Architecture Search (NAS) and Hyperparameter Optimization (HPO).** Deep learning derives its success from the manually designed feature extractors which automate the feature engineering process. Neural architecture search (NAS) (Elsken et al., 2019), on the other hand, aims at automating the very design of network architectures for a task at hand. NAS can be seen as a subset of HPO (Feurer & Hutter, 2019), which refers to the automated search for optimal hyperparameters, such as learning rate, batch size, dropout, loss function, optimizer, and architectural choices. Rapid and extensive research on NAS for image classification and object detection has been witnessed as of late (Liu et al., 2018; Zela et al., 2019; Xu et al., 2019; Pham et al., 2018; Cai et al., 2018). Deploying NAS techniques in face recognition systems has also seen a growing interest (Zhu, 2019; Wang, 2021). For example, reinforcement learning-based NAS strategies (Xu et al., 2019) and one-shot NAS methods (Wang, 2021) have been deployed to search for an efficient architecture for face recognition with low error. However, in a majority of these methods, the training hyperparameters for the architectures are *fixed*, which we observe should be reconsidered in order to obtain the fairest possible face recognition systems.

A few works have applied hyperparameter optimization to mitigate bias in models for tabular datasets. Perrone et al. (2021) recently introduced a Bayesian optimization framework to optimize accuracy of models while satisfying a bias constraint. The concurrent works of Schmucker et al. (2020) and Cruz et al. (2020) extend Hyperband (Li et al., 2017) to the multi-objective setting and show its applications to fairness. The former was later extended to the asynchronous setting (Schmucker et al., 2021). Lin et al. (2022) proposes de-biasing face recognition models through model pruning. However, they consider just two architectures and just one set of hyperparameters. To the best of our knowledge, no prior work uses any AutoML technique (NAS, HPO, or joint NAS and HPO) to design fair face recognition models, and no prior work uses NAS to design fair models for any application.

## 3 A LARGE-SCALE ANALYSIS OF ARCHITECTURES AND FAIRNESS

In this section, we seek to address the following question: *are architectures and hyperparameters important for fairness?* To this end, we conduct an exploration of many different model architectures using different hyperparameter combinations. We find strong evidence that accuracy is not predictive

of fairness metrics, which provides strong motivation for using NAS techniques to optimize fairness and accuracy *jointly*, which we explore in Section 4.

**Experimental Setup.** We train and evaluate each model configuration on a gender-balanced subset of the CelebA dataset (Liu et al., 2015). CelebA is a large-scale face attributes dataset with more than 200K celebrity images and a total of 10 177 gender-labeled identities. The dataset distribution has about 60% images labeled females and 40% labeled males. While this work analyzes phenotypic metadata (perceived gender), the reader should not interpret our findings absent a social lens of what these demographic groups mean inside society. We guide the reader to Hamidi et al. (2018) and Keyes (2018) for a look at these concepts for gender.

We use a balanced set of male and female identities, as is common practice in fairness research in face recognition work (Cherepanova et al., 2021b; Zhang & Deng, 2020). To study the importance of architectures and hyperparamters for fairness, we use the following training pipeline – ultimately conducting 355 training runs with different combinations of 29 architectures from the Pytorch Image Model (timm) database (Wightman, 2019) and hyperparameters. We conduct training runs with both the default hyperparameters as well as hyperparameters which are standardized across all architecutres, e.g., AdamW with lr=0.001 and SGD with lr=0.1. For each model, we use the default learning rate and optimizer that was published with that model. We then conduct a training run with these hyperparameters and each of three heads, ArcFace (Wang et al., 2018), CosFace (Deng et al., 2019), and MagFace (Meng et al., 2021). Next, we use that default learning rate with both AdamW (Loshchilov & Hutter, 2019) and SGD optimizers (again with each head). Finally, we also conduct training routines with AdamW and SGD with unifed learning rates (SGD with lr=0.1 and AdamW with lr=0.001). In total, we run a single architecture between 9 and 13 times (9 times if the default optimizer and learning rates were the same as the standardized, and 13 times otherwise). All other hyperparameters were the same for each model training run.

**Evaluation procedure.** When evaluating the performance of our models, we choose the standard approach in face identification tasks to evaluate the performance of the learned representations. Recall that face recognition models usually learn representations with an image backbone and then learn a mapping from those representations onto identities of individuals with the head of the model. As is commonplace (Cherepanova et al., 2021a; 2022), evaluating the learned feature representations allows us to better isolate the impact of the image backbone architecture and transfer this learned feature extractor onto other datasets (see Section 4.3.1).

The main performance metric for the models will be representation error, which we will henceforth simply refer to as *Error*. Recall that we pass each test image through a trained model and save the learned representation. To compute *Error*, we merely ask, for a given probe image/identity, whether the closest image in feature space is *not* of the same person based on $l_2$ distance.

We use a common fairness metric in face recognition which is explored in the NIST FRVT (Grother et al., 2010) and which we call *rank disparity*. To compute the rank of a given image/identity, we ask how many images of a different identity are closer to the image in feature space. We define this index as the *Rank* of a given image under consideration. Thus, *Rank(image)* = 0 if and only if *Error(image)* = 0; *Rank(image)* > 0 if and only if *Error(image)* = 1. We examine the **rank disparity** which is the absolute difference of the average ranks for each perceived gender in a dataset $\mathcal{D}$:

$$\text{Rank Disparity} = \left| \frac{1}{|\mathcal{D}_{\text{male}}|} \sum_{x \in \mathcal{D}_{\text{male}}} \text{Rank}(x) - \frac{1}{|\mathcal{D}_{\text{female}}|} \sum_{x \in \mathcal{D}_{\text{female}}} \text{Rank}(x) \right|.$$

We focus on rank disparity throughout this section and Section 4, and we explore other forms of fairness metrics in face recognition in Appendix A.3.

**Results and Discussion.** By plotting the performance of each training run with the error on the $x$-axis and rank disparity on the $y$-axis in Figure 2, we can easily conclude two main points. First, optimizing for error does not also optimize for fairness, and second, different architectures have different fairness properties.

On the first point, a search for architectures and hyperparameters which have high performance on traditional metrics does not translate to high performance on fairness metrics. We see that within

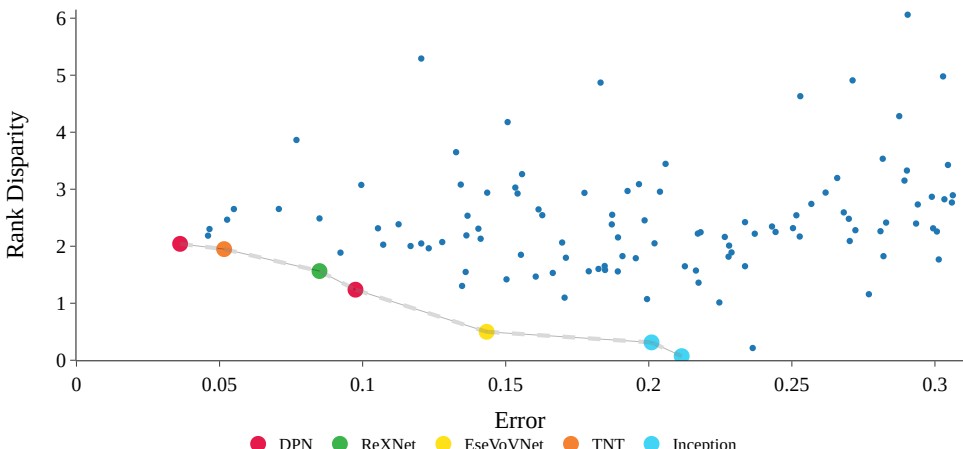

Figure 2: Error-Rank Disparity Pareto front of the architectures with lowest error (< 0.3). Models in the lower left corner are better. The Pareto front is notated with a dashed line. Other points are architecture and hyperparameter combinations which are not Pareto-optimal. DPN, ReXNet, EseVovNet, TNT, and Inception architectures are Pareto-optimal.

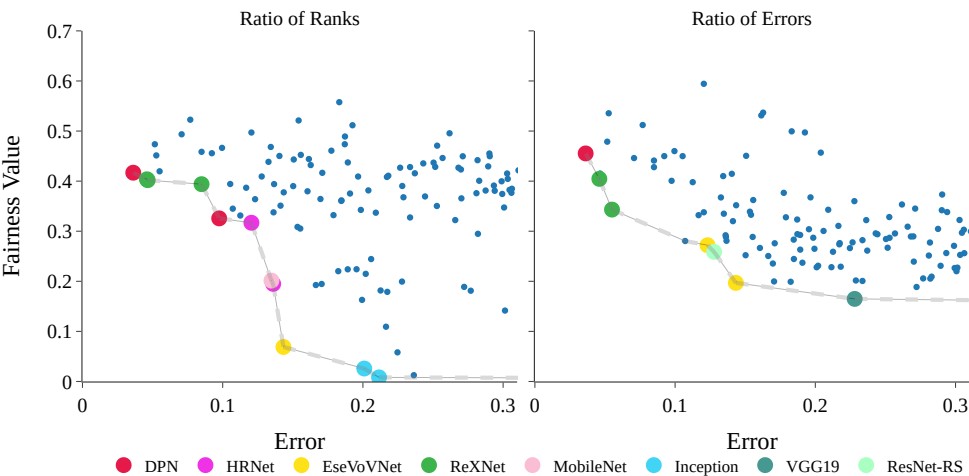

Figure 3: Depending on the fairness metric, different architectures are Pareto-optimal. On the left, we plot the metric Ratio of Ranks which admit DPN, ReXNet, HRNet, MobileNet, EseVovNet, and Inceptions as Pareto-optimal. On the right, we plot the metric Ratio of Errors where DPN, ReXNet, EseVovNet, ResNet-RS, and VGG19 are Pareto-optimal.

models with lowest error – those models which are most interesting to the community – there is low correlation between error and rank disparity ($\rho = -.113$ for models with error < 0.3). In Figure 2, we see that Pareto optimal models are versions of DPN, TNT, ReXNet, VovNet, and ResNets (in increasing error and decreasing fairness). We conclude that both architectures and hyperparameters play a significant role in determining the accuracy and fairness trade-off, motivating their joint optimization in Section 4.

Additionally, we observe that the Pareto curve is dependent upon what fairness metric we consider. For example, in Figure 3, we demonstrate that a very different set of architectures are Pareto optimal if instead of rank disparity (rank difference between perceived genders) we consider the ratio of ranks

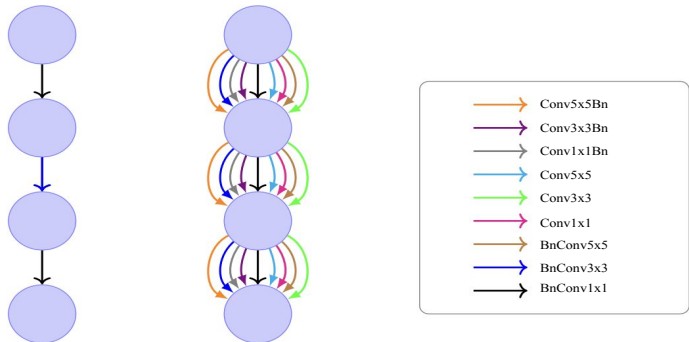

Figure 4: DPN block (left) vs. our searchable block (right).

Table 1: Searchable hyperparameter choices.

| Hyperparameter | Choices |
|---|---|
| Architecture Head/Loss | MagFace, ArcFace, CosFace |
| Optimizer Type | Adam, AdamW, SGD |
| Learning rate (conditional) | Adam/AdamW $\rightarrow [1e-4, 1e-2]$, SGD $\rightarrow [0.09, 0.8]$ |

between the two perceived genders or the ratio of the errors. Specifically, on the ratio of ranks metric, the Pareto frontier contains versions of HRNet, MobileNet, VovNet, and ResNet whereas the Pareto frontier under the ratio of errors metric includes versions of NesT, ResNet-RS, and VGG19.

Further, different architectures exhibit different optimal hyperparameters. For example, the Xception65 architecture finds SGD with ArcFace and AdamW with ArcFace are Pareto-optimal whereas the Inception-ResNet architecture finds MagFace and CosFace optimal with SGD. This illustrates the care that needs to be taken when choosing a model – optimizing architectures and hyperparameters for error alone will not lead to fair models.

Finally, existing architectures and hyperparameters do not yield models which simultaneously exhibit both low error and low disparity. For example, in Figure 2 there is a significant area under the Pareto curve. While there are models with very low error, in order to improve the disparity metric, one must sacrifice significant performance. However, in Section 4, we will see that our joint NAS+HPO experiments for rank disparity ultimately find a model convincingly in the area to the left of this Pareto curve – that is, we find a model with low error *and* disparity.

## 4  JOINT NAS+HPO FOR FAIRNESS

In this section, we employ joint NAS+HPO to find better architectures. Inspired by our findings on the importance of architecture and hyperparameters for fairness in Section 3, we initiate the first joint NAS+HPO study for fairness in face recognition. We start by describing our search space and search strategy. We then present a comparison between the architectures obtained from multi-objective joint NAS+HPO and the handcrafted image classification models studied in Section 3. We conclude that our joint NAS+HPO indeed discovers simultaneously accurate and fair architectures.

### 4.1  SEARCH SPACE DESIGN

We design our search space based on our analysis in Section 3. In particular, our search space is inspired by Dual Path Networks (Chen et al., 2017) due to its strong trade-off between rank disparity and accuracy as seen in Figure 2.

**Hyperparameter Search Space Design.**  We choose three categories of hyperparameters for NAS+HPO: the architecture head/loss, the optimizer, and the learning rate, depicted in Table 1.

**Architecture Search Space Design.** Dual Path Networks (Chen et al., 2017) for image classification share common features (ResNets (He et al., 2016a)) while possessing the flexibility to explore new features (Huang et al., 2017) through a dual path architecture. We replace the repeating `1x1_conv-3x3_conv-1x1_conv` block with a simple recurring searchable block depicted in Figure 4. Furthermore, we stack multiple such searched blocks to closely follow the architecture of Dual Path Networks. We have nine possible choices for each of the three operations in the DPN block as depicted in Table 2. The choices include a vanilla convolution, a convolution with pre-normalization and a convolution with post-normalization.

To summarize, our search space consists of a choice among 81 different architecture types, 3 different head types, 3 different optimizers, and a possibly infinite number of choices for the learning rate.

## 4.2 SEARCH STRATEGY

We navigate the search space using Black-Box-Optimization (BBO) with the following desiderata:

**Multi-fidelity optimization.** A single function evaluation for our use-case corresponds to training a deep neural network on a given dataset. This is generally quite expensive for traditional deep neural networks on moderately large datasets. Hence we would like to use cheaper approximations to speed up the black-box algorithm with multi-fidelity optimization techniques (Schmucker et al., 2021; Li et al., 2017; Falkner et al., 2018), e.g., by evaluating many configurations based on short runs with few epochs and only investing more resources into the better-performing ones.

**Multi-objective optimization.** We want to observe a trade-off between the accuracy of the face recognition system and the fairness objective of choice, so our joint NAS+HPO algorithm supports multi-objective optimization (Paria et al., 2020; Davins-Valldaura et al., 2017; Mao-Guo et al., 2009).

The SMAC3 package (Lindauer et al., 2022) offers a robust and flexible framework for Bayesian Optimization with few evaluations. SMAC3 offers a SMAC4MF facade for *multi-fidelity optimization* to use cheaper approximations for expensive deep learning tasks like ours. We choose ASHA (Schmucker et al., 2021) for cheaper approximations with the initial and maximum fidelities set to 25 and 100 epochs, respectively, and $\eta = 2$. Every architecture-hyperparameter configuration evaluation is trained using the same training pipeline as in Section 3. For the sake of simplicity, we use the ParEGO (Davins-Valldaura et al., 2017) algorithm for *multi-objective optimization* with $\rho$ set to 0.05.

## 4.3 RESULTS

We follow the evaluation scheme of Section 3 to compare the models discovered by joint NAS+HPO with the handcrafted image classification models. In Figure 5, we compare the set of models discovered by joint NAS+HPO vs. the models on the Pareto front studied in Section 3. We train each of these models for 4 seeds to study the robustness of error and disparity for the models. As seen in Figure 5, we Pareto-dominate all other models with above random accuracy on the validation set. On the test set, we still Pareto-dominate all highly competitive models (with $Error < 0.1$), but due to differences between the two dataset splits, one of the original configurations (DPN with Magface) also becomes Pareto-optimal. However, the error of this architecture is 0.13, which is significantly higher than the the best original model (0.05) and the SMAC models (0.03-0.04). Furthermore, from Figure 5 it is also apparent that some models such as VoVNet and DenseNet show very large standard errors across seeds. Hence, it becomes very important to also study the robustness of the models across seeds along with the accuracy and disparity Pareto front. We also compare to the current state of the art baseline ArcFace (Deng et al., 2019), which, using our training pipeline on CelebA data with face identification as our task, achieves an error of 4.35%. We however, outperform this architecture with our best performing novel architecture achieving an error of 3.10%.

In the work of Cherepanova et al. (2021b), they studied bias mitigation techniques for face recognition for the state of the art ArcFace models. The best technique from this work achieves an accuracy on males of 93.% and accuracy on females of 89.1% with a performance gap of 4.3%. Our novel architecture achieves accuracies of 96-98% on both males and females which means that our technique outperforms those reported in Cherepanova et al. (2021b) by a significant margin.

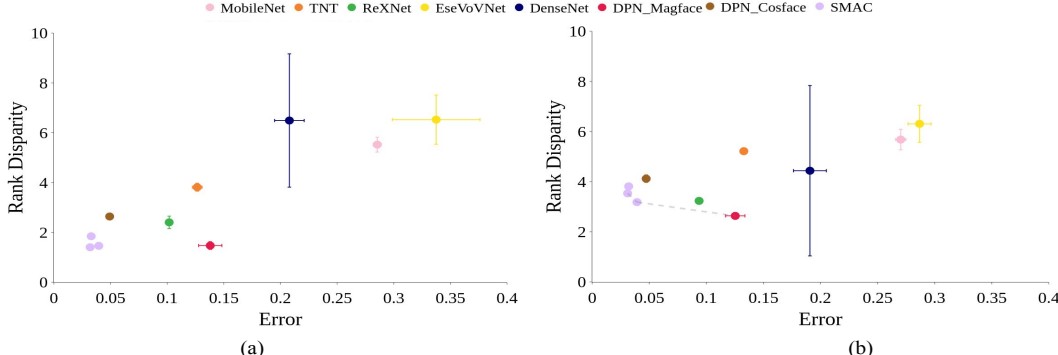

Figure 5: Pareto front of the models discovered by SMAC and the rank-1 models from `timm` for the *(a)* validation and *(b)* test sets on CelebA. Each point corresponds to the mean and standard error of an architecture after training for 4 seeds. The SMAC models Pareto-dominate the top performing `timm` models ($Error < 0.1$).

### 4.3.1 TRANSFER ACROSS FACE RECOGNITION DATASETS

Inspired by our findings on the CelebA dataset, we now study the accuracy-disparity trade-off of the models studied in Section 3 and the searched models from Section 4 on two different datasets. The first face recognition dataset we use is VGGFace2 (Cao et al., 2018), which is based on the same protected attribute (perceived gender) that has served as the focus of our study. The second dataset, Racial Faces in the Wild (RFW) (Wang et al., 2019a), consists of four different racial identities: Caucasian, Indian, Asian, and African. We compute the rank disparity within different *ethnicities*, i.e., a different attribute than the *perceived gender* studied in previous sections. With this dataset, we aim to study the generalization of the fair representations learned by the models across a different protected attribute. However, we caution the reader that the labels of these datasets rely on socially constructed concepts of gender presentation and ethnicity. The intention here is to study how the models discovered by SMAC generalize to these datasets and compare to the other handcrafted `timm` (Wightman, 2019) architectures.

To evaluate our models on these datasets, we directly transfer our models to the two test sets. That is, we use the models trained on CelebA, without re-training or fine-tuning the models on the new datasets. As observed in Figure 6, the models discovered using joint NAS+HPO still remain Pareto-optimal on both datasets. In the case of VGGFace2, the models found by SMAC are the only ones to have an error below 0.5, where the next-best model has an error above 0.7. In the case of RFW, the models found by SMAC have considerably lower rank disparity *and* error than the standard models studied in Section 3. This might be due to the optimized architectures learning representations that are intrinsically fairer than those of standard architectures, but it requires further study to test this hypothesis and determine in precisely which characteristics these architectures differ.

Additionally, we trained our novel architectures on VGGFace2 and find that we outperform the ArcFace baseline with both error and fairness metrics; See Table 5. We observe that while the ArcFace model has low error, it has high disparity, whereas our model with best error is a 36% increase in performance over the ArcFace Baseline and an 85% improvement in fairness. We also see that our best model by Rank Disparity, is not significantly worse than ArcFace, and we have a model that achieves similar performance to the ArcFace baseline and yields a 91% improvement in fairness.

Finally, when we test our trained models (on CelebA) on the test sets of other common face recognition datasets, our newly found SMAC models outperform the competitors in all cases. See Table 4.

## 5 CONCLUSION, FUTURE WORK AND LIMITATIONS

We conducted the first large-scale analysis of the relationship among hyperparameters and architectural properties, and accuracy, bias, and disparity in predictions. We trained a set of 29 architectures totalling 355 models and 88 493 GPU hours across different loss functions and architecture heads on

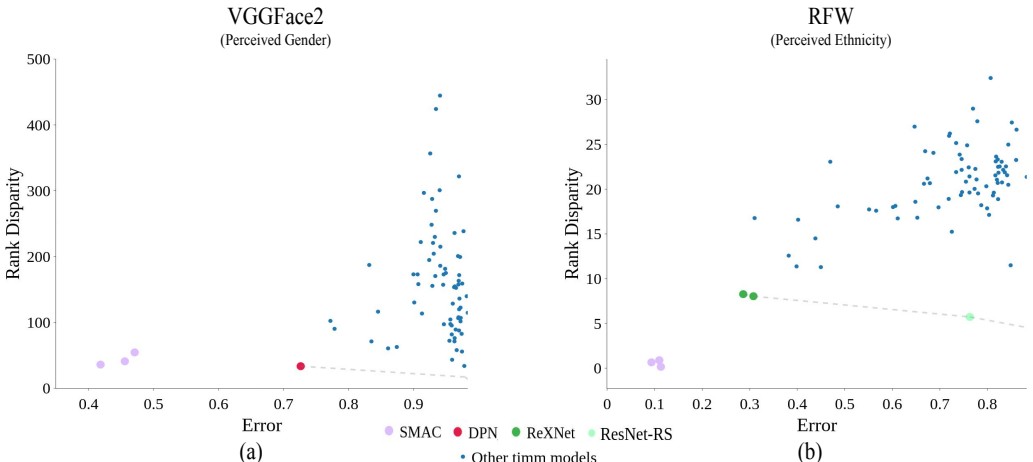

Figure 6: Pareto front of the models on *(a)* the VGGFace2 test set with perceived gender as the protected attribute and *(b)* the RFW test set with perceived ethnicity as the protected attribute. The SMAC models discovered by joint NAS+HPO Pareto-dominate the `timm` models.

CelebA face recognition, analyzing their inductive biases for fairness and accuracy. We also initiated the study of neural architecture search (NAS) for fairness. We constructed a search space based on the best architectures from the initial analysis, and, based on using SMAC3 for joint NAS+HPO, release a set of architectures that Pareto-dominate the most accurate models with respect to accuracy and rank disparity. By releasing all of our code and raw results, users can repeat all of our analyses and experiments with their fairness metric of interest.

**Future Work.** Since our work lays the foundation for studying NAS+HPO for fairness in face recognition, it opens up a plethora of opportunities for future work. We expect the future work in this direction to focus on studying different multi-objective algorithms (Fu & Liu, 2019; Laumanns & Ocenasek, 2002) and NAS techniques (Liu et al., 2018; Zela et al., 2019; White et al., 2021) to search for inherently fairer models. Further, it would be interesting to study how the properties of the architectures discovered translate across different demographics and populations. Another potential direction of future work is including priors and beliefs about fairness in the society from experts to further improve and aid NAS+HPO methods for fairness by integrating expert knowledge. Finally, given the societal importance of fairness, it would be interesting to study how our findings translate to real-life face recognition systems under deployment.

**Limitations.** While our work is a step forward in both studying the relationship among architectures, hyperparameters, and bias, and in using NAS techniques to mitigate bias in face recognition models, there are important limitations to keep in mind. Since we have studied our findings on only a few datasets, these may not generalize to other datasets and fairness metrics. Secondly, since face recognition applications span government surveillance (Hill, 2020b), target identification from drones (Marson & Forrest, 2021), and identification in personal photo repositories (Google, 2021), our findings need to be studied thoroughly across different demographics before they could be deployed in real-life face recognition systems. Further, it is important to consider how the mathematical notions of fairness used in research translate to those actually impacted (Saha et al., 2020), which is a broad concept without a concise definition. Before deploying a particular system that is meant to improve fairness in a real-life application, we should always critically ask ourselves whether doing so would indeed prove beneficial to those impacted by the given sociotechnical system under consideration or whether it falls into one of the traps described by Selbst et al. (2019). In contrast to some other works, we do, however, feel, that our work helps to overcome the portability trap since it empowers domain experts to optimize for the right fairness metric, in connection with public policy experts, for the problem at hand rather than only narrowly optimizing one specific metric.

## 6 ETHICS STATEMENT

Face recognition systems are being used for more and more parts of daily lives, from government surveillance (Hill, 2020b), to target identification from drones (Marson & Forrest, 2021), to identification in personal photo repositories (Google, 2021). It is also increasingly evident that many of these models are biased based on race and gender (Grother et al., 2019; Raji et al., 2020; Raji & Fried, 2021). If left unchecked, these technologies, which make biased decision for life-changing events, will only deepen existing societal harms. Our work seeks to better understand and mitigate the negative effects that biased face recognition models have on society. By conducting the first large-scale study of the effect of architectures and hyperparameters on bias, and by developing and open-sourcing face recognition models that are more fair than all other competitive models, we provide a resource for practitioners to understand inequalities inherent in face recognition systems and ultimately advance fundamental understandings of the harms and technological ills of these systems.

That said, we would like to address potential ethical challenges of our work. We believe that the main ethical challenge of this work centers on our use of certain datasets. We acknowledge that the common academic datasets which we used to evaluate our research questions, CelebA (Liu et al., 2015), VGGFace2 (Cao et al., 2018), and RFW (Wang et al., 2019b), are all datasets of images scraped from the web without the informed consent of those whom are depicted. This ethical challenge is one that has plagued the research and computer vision community for the last decade (Peng et al., 2021; Paullada et al., 2021) and we are excited to see datasets being released which have fully informed consent of the subjects, such as the Casual Conversations Dataset (Hazirbas et al., 2021). Unfortunately, this dataset in particular has a rather restrictive license, much more restrictive than similar datasets, which prohibited its use in our study.

We also acknowledge that while our study is intended to be constructive in performing the first neural architecture search experiments with fairness considerations, the specific ethical challenge we highlight is that of unequal or unfair treatment by the technologies. We note that our work could be taken as a litmus test which could lead to the further proliferation of facial recognition technology which could cause other harms. If a system demonstrates that it is less biased than other systems, this could be used as a reason for the further deployment of facial technologies and could further impinge upon unwitting individual's freedoms and perpetuate other technological harms.

Experiments were conducted using a private infrastructure, which has a carbon efficiency of 0.373 $kgCO_2eq/kWh$. A cumulative of 88 493 hours of computation was performed on hardware of type RTX 2080 Ti (TDP of 250W). Total emissions are estimated to be 8,251.97 $kgCO_2eq$ of which 0% was directly offset. Estimations were conducted using the MachineLearning Impact calculator presented in Lacoste et al. (2019). By releasing all of our raw results, code, and models, we hope that our results will be widely beneficial to researchers and practitioners with respect to designing fair face recognition systems.

## 7 REPRODUCIBILITY STATEMENT

We ensure that all of our experiments are reproducible by releasing our code and raw data files at `https://anonymous.4open.science/r/FR-NAS-92EC`. We also release the instructions to reproduce our results with the code. Furthermore, we release all of the configuration files for all of the models trained. Our experimental setup is described in Section 3 and Appendix A.1. We provide clear documentation on the installation and system requirements in order to reproduce our work. This includes information about the computing environment, package requirements, dataset download procedures, and license information. We have independently verified that the experimental framework is reproducible which should make our work and results and experiments easily accessible to future researchers and the community.

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

# A    FURTHER DETAILS ON EXPERIMENTAL DESIGN AND RESULTS

## A.1    EXPERIMENTAL SETUP

The list of the models we study from `timm` are: `coat_lite_small` (Xu et al., 2021), `convit_base` (d'Ascoli et al., 2021), `cspdarknet53` (Wang et al., 2020a), `dla102x2` (Yu et al., 2018), `dpn107` (Chen et al., 2017), `ese_vovnet39b` (Lee & Park, 2020), `fbnetv3_g` (Dai et al., 2021), `ghostnet_100` (Han et al., 2020b), `gluon_inception_v3` (Szegedy et al., 2016), `gluon_xception65` (Chollet, 2017), `hrnet_w64` (Sun et al., 2019), `ig_resnext101_32x8d` (Xie et al., 2016), `inception_resnet_v2` (Szegedy et al., 2017), `inception_v4` (Szegedy et al., 2017), `jx_nest_base` (Zhang et al., 2021), `legacy_senet154` (Hu et al., 2018), `mobilenetv3_large_100` (Howard et al., 2019), `resnetrs101` (Bello et al., 2021), `rexnet_200` (Han et al., 2020a), `selecsls60b` (Mehta et al., 2019), `swin_base_patch4_window7_224` (Liu et al., 2021), `tf_efficientnet_b7_ns'` (Tan & Le, 2019), `'tnt_s_patch16_224`(Han et al., 2021), `twins_svt_large` (Chu et al., 2021) , `vgg19` (Simonyan & Zisserman, 2014), `vgg19_bn` (Simonyan & Zisserman, 2014), `visformer_small` (Chen et al., 2021), `xception` and `xception65` (Chollet, 2017).

We study at most 13 configurations per model ie 1 default configuration corresponding to the original model hyperparameters with CosFace as head. Further, we have at most 12 configs consisting of the 3 heads (CosFace, ArcFace, MagFace) $\times$ 2 learning rates(0.1,0.001) $\times$ 2 optimizers (SGD, AdamW). All the other hyperparameters are held constant for training all the models. All model configurations are trained with a total batch size of 64 on 8 RTX2080 GPUS for 100 epochs each.

## A.2    OBTAINED ARCHITECTURES AND HYPERPARAMETER CONFIGURATIONS FROM BLACK-BOX-OPTIMIZATION

In Figure 7 we present the architectures and hyperparameters discovered by SMAC. Particularly we observe that `conv 3x3` followed `batch norm` is a preferred operation and CosFace is the preferred head/loss choice.

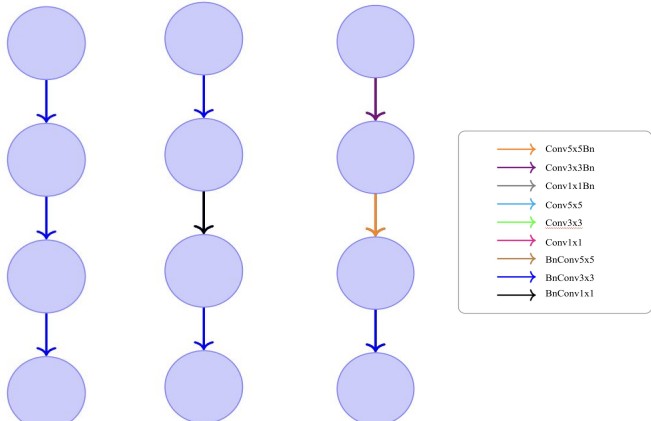

Figure 7: SMAC discovers the above building blocks with (a) corresponding to architecture with CosFace, with SGD optimizer and learning rate of 0.2813 as hyperparamters (b) corresponding to CosFace, with SGD as optimizer and learning rate of 0.32348 and (c) corresponding to CosFace, with AdamW as optimizer and learning rate of 0.0006

## A.3    ANALYSIS OF THE PARETO-FRONT OF DIFFERENT FAIRNESS METRICS

In this section, we include additional plots that support and expand on the main paper. Primarily, we provide further context of the Figures in the main body in two ways. First, we provide replication plots of the figures in the main body but for all models. Recall, the plots in the main body only show

Table 2: Operation choices and definitions.

| Operation | Definition |
|---|---|
| BnConv1x1 | Batch Normalization → Convolution with 1x1 kernel |
| Conv1x1Bn | Convolution with 1x1 kernel → Batch Normalization |
| Conv1x1 | Convolution with 1x1 kernel |
| BnConv3x3 | Batch Normalization → Convolution with 3x3 kernel |
| Conv3x3Bn | Convolution with 3x3 kernel → Batch Normalization |
| Conv3x3 | Convolution with 3x3 kernel |
| BnConv5x5 | Batch Normalization → Convolution with 5x5 kernel |
| Conv5x5Bn | Convolution with 5x5 kernel → Batch Normalization |
| Conv5x5 | Convolution with 5x5 kernel |

Table 3: Fairness Metrics Overview

| Fairness Metric | Equation |
|---|---|
| Disparity | $\lvert Accuracy(male) - Accuracy(female)\rvert$ |
| Rank Disparity | $\lvert Rank(male) - Rank(female)\rvert$ |
| Ratio | $\lvert 1 - \frac{Accuracy(male)}{Accuracy(female)}\rvert$ |
| Rank Ratio | $\lvert 1 - \frac{Rank(male)}{Rank(female)}\rvert$ |
| Error Ratio | $\lvert 1 - \frac{Error(male)}{Error(female)}\rvert$ |
| BPC (Dhar et al., 2021) | $\frac{Bias - Bias_{deb}}{Bias} - \frac{TPR - TPR_{deb}}{TPR}$ |
| StdAcc | $\lvert Std(Acc(male)) - Std(Acc(female))\rvert$ |

models with Error<0.3, since high performing models are the most of interest to the community. Second, we also show figures which depict other fairness metrics used in facial recognition. The formulas for these additional fairness metrics can be found in Table 3.

We replicate Figure 2 in Figure 8; Figure 3 in Figure 9; Figure 6 in Figure 10 and Figure 11. We add additional metrics with Disparity being plotted in Figure 12, Ratio being plotted in Figure 13, BPC being plotted in Figure 14, and StdAcc being plotted in Figure 15.

## A.4 EVALUATION ON BENCHMARKS

We further evaluate our models pre-trained on CelebA on different face recognition benchmarks (without fine-tuning) Table 4. We observe that SMAC models are the best or second best for all the benchmarks.

## A.5 TRAINING ON VGGFACE2

We trained our novel architectures on VGGFace-2 and find that we outperform the ArcFace baseline with both error and fairness metrics; See Table 5. We observe that while the ArcFace model has low error, it has high disparity, whereas our model with best error (SMAC_680) is a 36% increase in performance over the ArcFace Baseline and an 85% improvement in fairness. We also see that our best model by Rank Disparity (SMAC_101, is not significantly worse than ArcFace, and we have a model (SMAC_000) that achieves similar performance to the ArcFace baseline and yields a 91% improvement in fairness.

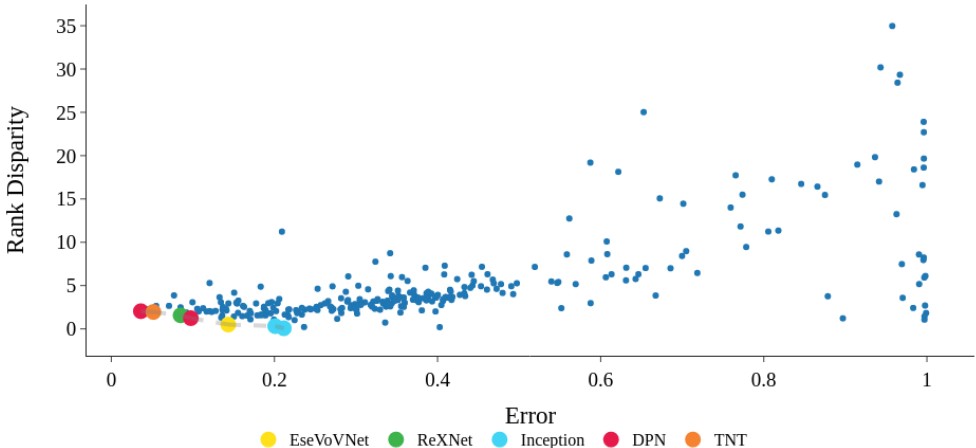

Figure 8: Replication of Figure 2 with all data points. Error-Rank Disparity Pareto front of the architectures with any non-trivial error. Models in the lower left corner are better. The Pareto front is notated with a dashed line. Other points are architecture and hyperparameter combinations which are not Pareto-dominant. DPN, ReXNet, EseVovNet, TNT, and Inception architectures are Pareto-dominant.

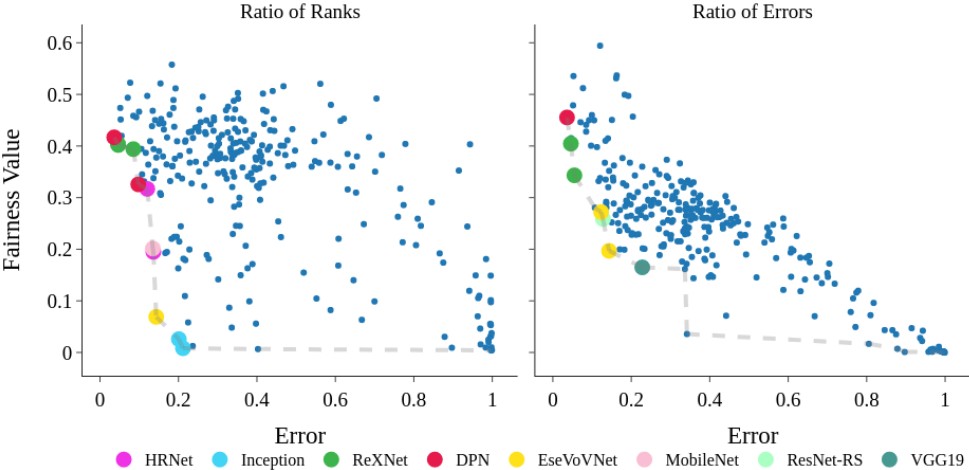

Figure 9: Replication of Figure 3 with all data points. Depending on the fairness metric, different architectures are Pareto-optimal. On the left, we plot the metric Ratio of Ranks which admit DPN, ReXNet, HRNet, MobileNet, EseVovNet, and Inceptions as Pareto-optimal. On the right, we plot the metric Ratio of Errors where DPN, ReXNet, EseVovNet, ResNet-RS, and VGG19 are architectures which are Parto-optimal

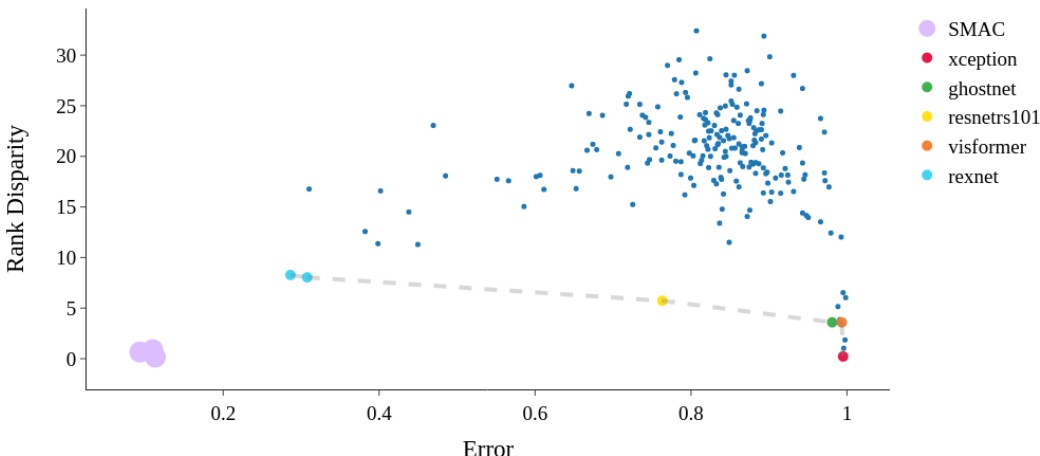

Figure 10: Replication of Figure 6 for VGGFace2 with all data points. Pareto front of the models on VGGFace2 test set with perceived gender as the protected attribute. The SMAC models discovered by joint NAS and HPO Pareto-dominate the `timm` models

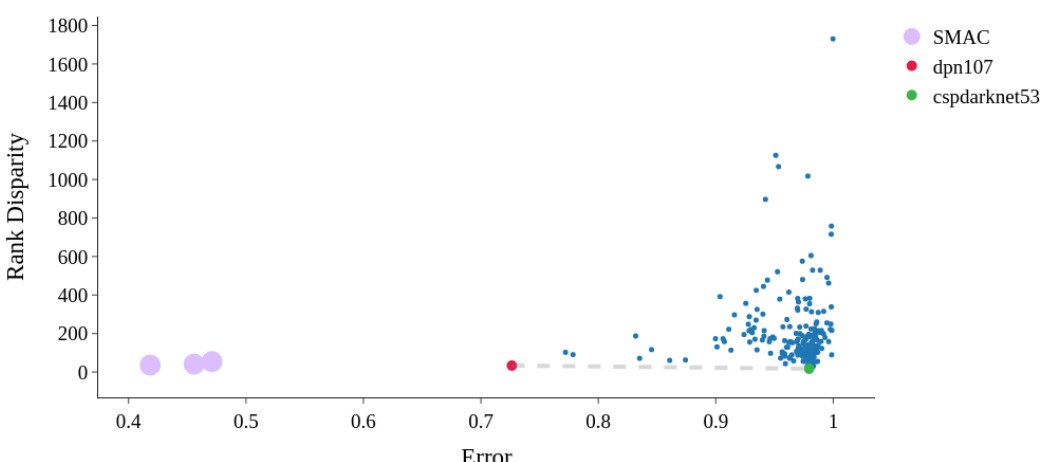

Figure 11: Replication of Figure 6 for RFW with all data points. Pareto front of the models on RFW test set with perceived ethnicity as the protected attribute. The SMAC models discovered by joint NAS and HPO Pareto-dominate the `timm` models

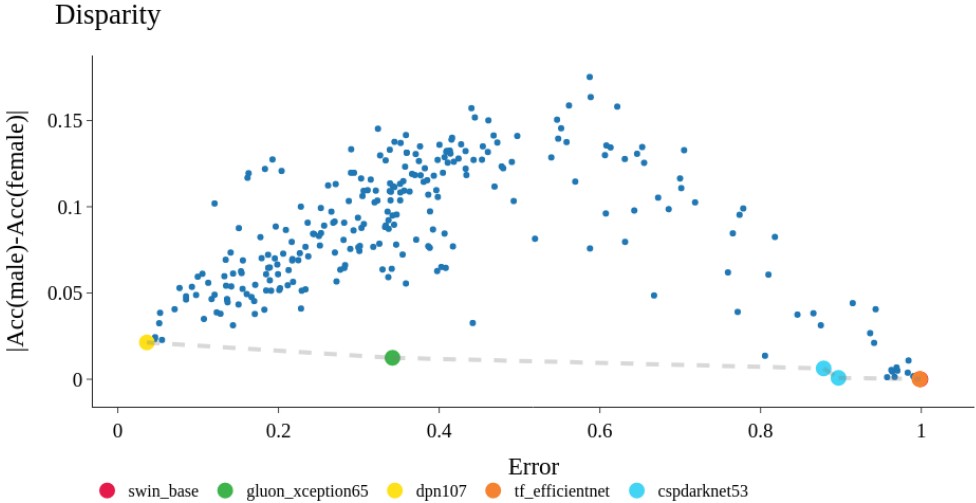

Figure 12: Extension of Figure 2 with all data points with the Disparity in accuracy metric. Error-Disparity Pareto front of the architectures with any non-trivial error. Models in the lower left corner are better. The Pareto front is notated with a dashed line. Other points are architecture and hyperparameter combinations which are not Pareto-dominant.

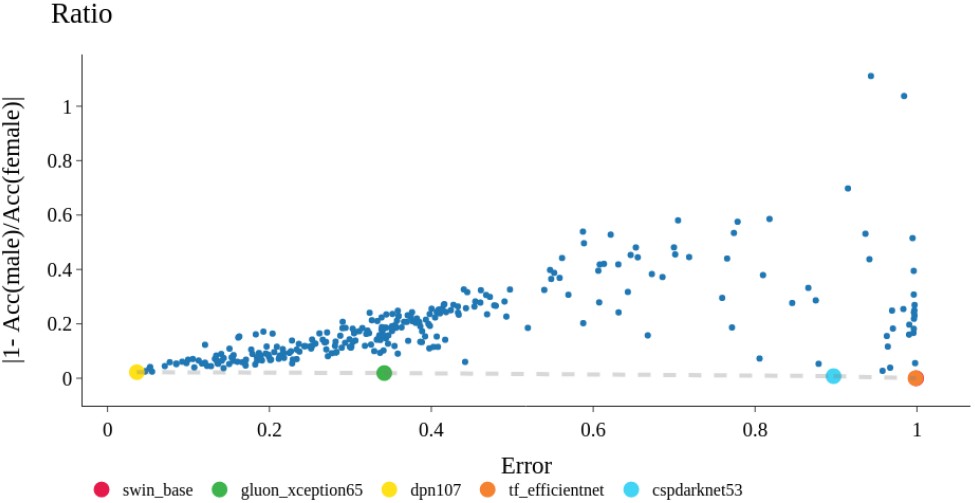

Figure 13: Extension of Figure 2 with all data points with the Ratio in accuracy metric. Error-Ratio Pareto front of the architectures with any non-trivial error. Models in the lower left corner are better. The Pareto front is notated with a dashed line. Other points are architecture and hyperparameter combinations which are not Pareto-dominant.

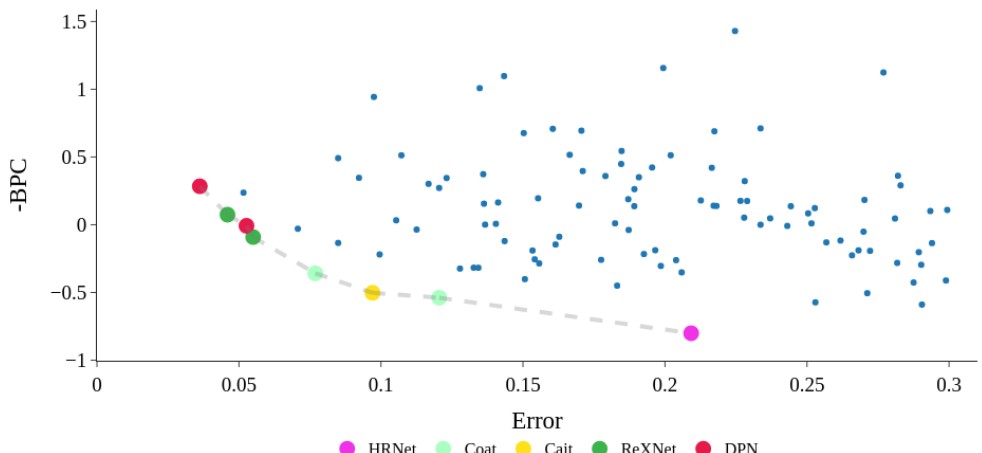

Figure 14: Extension of Figure 2 with all data points with the BPC metric. Error-BPC Pareto front of the architectures with any non-trivial error. Models in the lower left corner are better. The Pareto front is notated with a dashed line. Other points are architecture and hyperparameter combinations which are not Pareto-dominant.

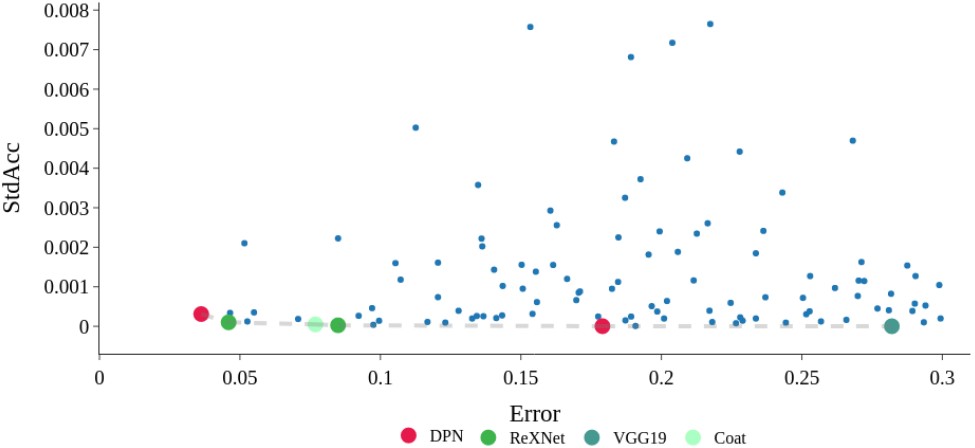

Figure 15: Extension of Figure 2 with all data points with the StdAcc metric. Error-StdAcc Pareto front of the architectures with any non-trivial error. Models in the lower left corner are better. The Pareto front is notated with a dashed line. Other points are architecture and hyperparameter combinations which are not Pareto-dominant.

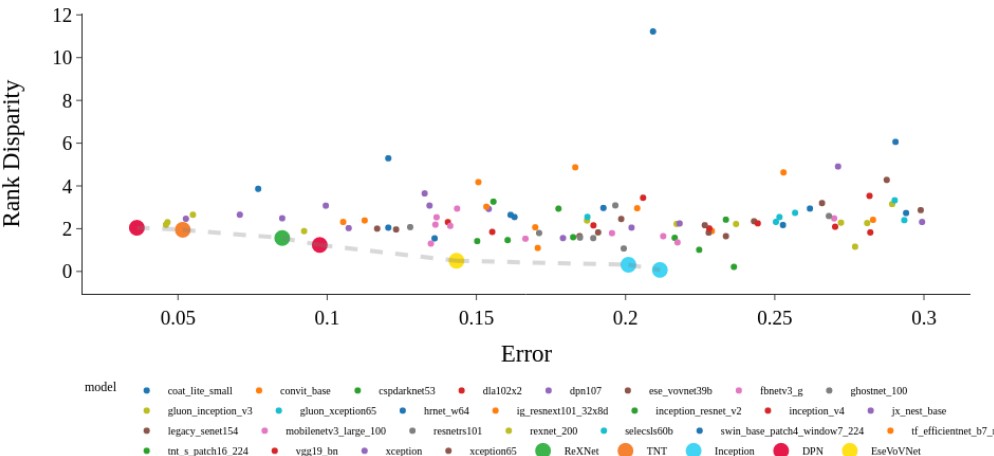

Figure 16: Replication of Figure Figure 8 with each architecture type represented by colors. Error-Rank Disparity Pareto front of the architectures with lowest error (< 0.3). Models in the lower left corner are better. The Pareto front is notated with a dashed line. Other points are architecture and hyperparameter combinations which are not Pareto-optimal. DPN, ReXNet, EseVovNet, TNT, and Inception architectures are Pareto-optimal.

Table 4: Evaluations on Face Recognition Benchmarks. The best accuracies are highlighted in red and the second best in blue.

| Architecture(trained on CelebA) | LFW | CFP_FF | CFP_FP | AgeDB | CALFW | CPLFW |
|---|---|---|---|---|---|---|
| Rexnet_200 | 71.18 | 73.62 | 54.07 | 56.31 | 61.01 | 52.22 |
| DPN_CosFace | 88.86 | 90.47 | 68.53 | 64.84 | 76.09 | 60.66 |
| DPN_MagFace | 85.88 | 89.03 | 61.30 | 60.00 | 73.50 | 55.53 |
| DenseNet161 | 81.72 | 81.88 | 64.82 | 55.16 | 65.7 | 58.40 |
| Ese_Vovnet39b | 73.31 | 74.42 | 63.33 | 50.00 | 59.86 | 57.93 |
| ArcFace | 73.36 | 76.30 | 62.64 | 57.41 | 63.62 | 57.66 |
| SMAC_000 | 94.98 | 95.60 | 74.24 | 80.23 | 84.73 | 64.22 |
| SMAC_010 | 94.22 | 95.08 | 75.14 | 82.35 | 85.35 | 66.26 |
| SMAC_680 | 87.45 | 90.34 | 64.22 | 61.28 | 76.16 | 56.16 |

Table 5: Comparing our model to ArcFace baselines when trained on VGGFace2 dataset. The best performance is highlighted in red and the second best in blue.

| Architecture | Error | Rank Disparity |
|---|---|---|
| ArcFace | 4.50 | 10.08 |
| SMAC_000 | 4.70 | 0.86 |
| SMAC_010 | 6.77 | 0.76 |
| SMAC_680 | 2.89 | 1.54 |

