# OpenReview forum: "On the Importance of Architectures and Hyperparameters for Fairness in Face Recognition"
_ICLR.cc/2023/Conference — Submitted to ICLR 2023_

### Official Review · Reviewer_7yRH · 2022-10-23

**Confidence:** 3
**Correctness:** 3
**Technical Novelty And Significance:** 2
**Empirical Novelty And Significance:** 2
**Recommendation:** 5

**Clarity, Quality, Novelty And Reproducibility:**

Clarity: 8 out of 10. The writing is clear and easy to follow.
Quality: 6 out of 10. The engineering part of this paper is solid. The results are beautiful.
Novelty: 3 out of 10. Although this is the first paper to use NAS and HPO to enhance the fairness, from my perspective, it lacks technical novelty and does not provide new conclusion.
Reproducibility: 8 out of 10. Authors have included the code.

**Details Of Ethics Concerns:**

This paper has included a thoughtful Ethics Statement to clarify the potential problem in using techniques and discussed methods to response them.

**Strength And Weaknesses:**

Strength:
1. This paper conducts a thorough experiment to analyze the relation among the hyper-parameters and architectures, and the accuracy, bias and disparity.
2. This paper concerns fairness, a long-stand problem in face recognition.
3. The final results outperform other algorithms by a large margin.

Weaknesses:
1. The conclusion of the relation among the hyper-parameters, architectures and fairness is too natural to me. Actually, if you conduct this analysis on other tasks, like face detection, face alignment, face reconstruction, face synthesis, etc., I believe this conclusion still holds as well. This is why the tasks like NAS and HPO exist. So even though the efforts were really worth respecting in this part, I did not find the conclusion exciting.
2. In the NAS and HPO, from my perspective, authors mostly borrow existing algorithms, where I still fail to find much novelty.


**Summary Of The Paper:**

This paper derives a conclusion that the hyper-parameters and architectural properties are relative to the accuracy, bias, and disparity. Thus they conduct NAS and HPO to obtain a powerful architecture and hyper-parameters, which outperforms other methods in terms of accuracy and fairness by a large margin.

**Summary Of The Review:**

In conclusion, I think the project is a solid work, and the paper is a nice report, but lacks technical novelty to be a ICLR paper.

---

> ### Author Response · Authors · 2022-11-17
> **Response to reviewer 7yRH**
>
> Thank you for your positive comments on the clear writing of our paper, extensiveness of our experiments and identifying the long standing problem of using NAS and HPO for fairness which we target. We address each of your comments below:
>
> > Technical Novelty
>
> To the best of our knowledge, our work is the first to extensively study the effect of hyperparameters and architectures for fairness in face recognition. Our main contribution is that architecture has a big influence on fairness, and this can be exploited to discover more fair architectures. We believe that this is impactful towards enabling and encouraging future research to study how NAS and HPO to make models fairer in a wide range of applications.
>
> We agree with the reviewer that our paper does not introduce new methods, but we respectfully point out that making the raw results from our extensive studies available and initiating NAS and HPO for fairness are novel contributions that will be very useful to the machine learning community. We would also like to mention that experimental-analysis papers have been accepted to recent ICLR, ICML, and NeurIPS conferences, and some of them have public reviews and meta-reviews available here:
>
> [1] Yang et al., ICLR 2020. [NAS evaluation is frustratingly hard.](https://openreview.net/forum?id=HygrdpVKvr&noteId=C5rPJ2sLzP)
>
> [2] Schmidt et al., ICML 2021. [Descending through a Crowded Valley - Benchmarking Deep Learning Optimizers.](http://proceedings.mlr.press/v139/schmidt21a.html)
>
> [3] White et al., [NeurIPS 2021. How Powerful are Performance Predictors in Neural Architecture Search?](https://openreview.net/forum?id=6RB77-6-_oI&noteId=l5H13jujQKo)
>
> [4] Ning et al., [NeurIPS 2021. Evaluating Efficient Performance Estimators of Neural Architectures.](https://openreview.net/forum?id=Esd7tGH3Spl&noteId=dl91y63zePD)
>
> [5] Mehta et-al, ICLR 2022 [NAS-Bench-Suite: NAS Evaluation is (Now) Surprisingly Easy](https://openreview.net/forum?id=0DLwqQLmqV)
>
>
> > Conclusion not exciting enough
>
> Our work aims at establishing the following three key points:
> * Architectures and Hyperparameters matter for fairness in face recognition. (Section 3)
> * Joint NAS and HPO helps discover architectures which are better in terms of both fairness and accuracy
> * As the first work highlighting and investigating architectures and hyperparameters for fairness in face recognition, we aim to empower domain experts with a methodology to achieve a good tradeoff between fairness and accuracy using NAS and HPO.
>
> Given the above contributions, our work lays the cornerstone for future research on NAS/HPO for fairness, with extensive experiments and analysis to support our claims. This in our opinion is both impactful and valuable to the machine learning community.
>
> Thank you again for your thoughtful review. We made a significant effort to address each of your questions and would appreciate it if you would consider raising your score in light of our response. Please let us know if you have additional questions we can address.

---

> > ### Author Response · Authors · 2022-12-03
> > **Happy to engage in further discussion**
> >
> > Thank you again for your feedback! We have made a significant effort to address each of your points, including new experiments and paper edits, and we would appreciate it if you would consider raising your score. We are eager to hear and discuss any further feedback you have.

---

> ### Author Response · Authors · 2022-12-08
> **Follow up for additional engagement**
>
> As we come to the end of Discussion Stage 2, we are happy to clarify any additional points.  We have conducted extensive additional experiments, including ones which demonstrate [the superiority of our approach to existing bias mitigation strategies from the literature](https://openreview.net/forum?id=iiRDsy85uXi&noteId=Hn-FB0aH0Z), namely simply swapping out the architecture of a face rec system improves fairness metrics more so than specially designed fair training routines.  This result will likely impact the direction of future fairness research as the inductive biases of our models are evidently more influential than fairness interventions during training.  Our additional experiments also address each of the points raised by reviewers where appropriate as do the edits we made to our updated draft.  Reviewer pGLd has raised their score to an 8 in light of our response, and we have also responded to each other reviewer equally comprehensively.  Do you have any additional questions we can address?

---

### Official Review · Reviewer_pGLd · 2022-10-24

**Confidence:** 4
**Correctness:** 3
**Technical Novelty And Significance:** 3
**Empirical Novelty And Significance:** 3
**Recommendation:** 8

**Clarity, Quality, Novelty And Reproducibility:**

The paper is well-written and also provides the code for reproducibility purpose. However, since the paper is mainly an experimental paper of studying the correlation between the fairness and network architecture/hyperparameter, they employ the existing technologies for the job. Although it is of limited technology novelties, the finding of this work provide some references for the future research for the same topic.

**Details Of Ethics Concerns:**

The paper has the section addressing ethics issues since face data and face recognition are sensitive now. However, the paper mainly uses the publicly available dataset for study, and they focus on how to improve the fairness. Thus, there is no big issues about the work.

**Strength And Weaknesses:**

Strength:
1. The work runs a very large scale architecture, hyperparameter search, and training  for face recognition  to study the relation between the fairness metric and accuracy under different perceived attributes (gender and ethnicity).
2. Although this is an experimental paper without any novel methodologies, the thorough experimental results provide insight of how different architectures and hyperparameters could affect the performance of fairness.

Weakness:
1. The study is mainly conducted using the CelebA dataset which only contains around 200 thousands face images for around 10,000 people and most of the faces are in frontal pose. The dataset is relatively small with respect to commonly used MSCeleb-1M, VGGFace-2, Glint360K, or WebFace260M (unfortunately, the proposed approach does not take VGGFace2 for any training.) The performance study may not generalize well to other face datasets.
2. The fairness metrics proposed in other recent works are not considered together. For example, the bias/performance coefficient proposed in "Dhar, Prithviraj, Joshua Gleason, Aniket Roy, Carlos D. Castillo, and Rama Chellappa. "PASS: Protected Attribute Suppression System for Mitigating Bias in Face Recognition." In Proceedings of the IEEE/CVF International Conference on Computer Vision (ICCV), pp. 15087-15096. 2021."
The metric, rank disparity, used in the paper only considers the performance of face retrieval and cannot fully cover the performance of face verification which is usually measured by true positive rate vs false negative rate. This limits the usefulness of the conducted studies.


**Summary Of The Paper:**

Unlike other previous works, the work first studies the fairness issues for face recognition in terms of the neural architecture and hyperparameter search. The authors run a large-scale study of 29 architectures from ViT to Xception for 355 models in total with different hyperparameters and spend 88,493 GPU hours. They culminate in a set of architectures which pareto-dominate all models in a large set of modern architectures. The works provide a foundation for the future works to further study the fairness in terms of network architecture and hyperparameters.


**Summary Of The Review:**

My main concern about the paper is that the dataset for study may not be representative enough to reflect the performance of current state-of-the-art face recognition models and the metric studies in the paper could not well represent the trade-off between face recognition accuracy and fairness since the face recognition contains two main tasks, face identification and face verification. The rank metric seems to reflect the performance of face identification but ignore face verification. If the authors could well-addressed my concern, I will change my rating.

---

> ### Author Response · Authors · 2022-11-17
> **Response to reviewer pGLd**
>
> Thank you for your constructive feedback and for highlighting that our work provides a foundation for future works to further study the impact of architectures and hyperparameters for fairness in face recognition. We are glad that you find our experiments thorough and extensive. We address each of your comments below:
>
> > (unfortunately, the proposed approach does not take VGGFace2 for any training.) The performance study may not generalize well to other face datasets.
>
> Prompted by your suggestion, we have trained our novel architectures on VGGFace-2 and find that we outperform the ArcFace baseline with both error and fairness metrics. We observe that while the ArcFace model has low error, it has high disparity, whereas our model with best error (SMAC_680) is a 36% increase in performance over the ArcFace Baseline and an 85% improvement in fairness. We also see that our best model by Rank Disparity (SMAC_101), is not significantly worse than ArcFace, and we have a model (SMAC_000) that achieves similar performance to the ArcFace baseline and yields a 91% improvement in fairness.
> | Model                                                                                     |     Error |   Rank Disparity |
> |:------------------------------------------------------------------------------------------|----------:|-----------------:|
> | SMAC_000     | 4.70 |         0.86 |
> | SMAC_680  | **2.89** |         1.53   |
> | SMAC_010    | 6.77 |         **0.76** |
> | ArcFace  | 4.50  |         10.0776   |
>
> > Reporting the accuracy comparison between the searched network and SOTA results
>
> This is a very useful suggestion. We show that our architecture achieves similar or better performance than the current state of the art on face identification on these datasets. The state of the art performance for face recognition tasks broadly belongs to the ArcFace model. Using our training pipeline on CelebA data with face identification as our task, the ArcFace model achieves an error of 4.35%, and our best performing novel architecture achieves an error of 3.10%.
> In our additional experiments training on VGGFace2, we observe that the ArcFace state of the art is an error rate of 4.5% and our best performing novel architecture achieves an error of 2.89%.
>
> > The fairness metrics proposed in other recent works are not considered together.
>
> The fairness metric that we use, which we call Rank Disparity, is the same metric used in the NIST FRVT report on Demographic effects for face identification (FNIR/CRC) [1]. We agree that this metric is designed for face identification and not face verification, which is natural given how these are two different technologies under the facial recognition umbrella.
>
> While the BPC metric is used for verification in [2], we have implemented this for our study, replotting Figure 2 from our paper with the BPC metric and including it in the appendix as Figure 14. Interestingly, many of the same models which are Pareto-dominant with the Rank Disparity metric are also Pareto-dominant with BPC.
>
> [1] https://nvlpubs.nist.gov/nistpubs/ir/2019/NIST.IR.8280.pdf
>
> [2] Dhar, Prithviraj, Joshua Gleason, Aniket Roy, Carlos D. Castillo, and Rama Chellappa. ["PASS: Protected Attribute Suppression System for Mitigating Bias in Face Recognition." In Proceedings of the IEEE/CVF International Conference on Computer Vision (ICCV), pp. 15087-15096. 2021.](https://arxiv.org/pdf/2108.03764.pdf)
>
> Thank you again for your thoughtful review. We made a significant effort to address each of your questions and would appreciate it if you would consider raising your score in light of our response. Please let us know if you have additional questions we can address.

---

> > ### Comment · Reviewer_pGLd · 2022-12-02
> > **Reply to the author response**
> >
> > I think the authors have conducted significant efforts to provide additional results to answer our questions and revise the paper accordingly. With the revised manuscript and the additional results, I feel it is okay to consider this paper for acceptance than the previous version.

---

### Official Review · Reviewer_i8qM · 2022-10-25

**Confidence:** 3
**Correctness:** 3
**Technical Novelty And Significance:** 2
**Empirical Novelty And Significance:** 2
**Recommendation:** 5

**Clarity, Quality, Novelty And Reproducibility:**

The paper has provided enough evidence to prove that recognition accuracy improvements cannot lead to fairness. They have clearly stated their search methods and searching strategies. Since they released their code, it would be easy to reproduce their results. However, they just applied the existing NAS on face recognition in terms of accuracy and fairness. It may not be novel enough.

**Strength And Weaknesses:**

Strengths:
1, Extensive experiments on existing face recognition algorithms reported in the paper may help researchers better understand those baselines and their hyperparameters.
2, The code and raw result files are released, which makes it easier for readers to follow the work.
3, They outperform other structures in terms of accuracy and fairness on VGGFace2 and RFW datasets by conducting NAS and hyperparameter optimization.

Weakness:
1, The proposed method seems to need more novelty. There are no new concepts or search strategies presented.
2, Although the paper is focused on recognition fairness. It might be worth reporting the accuracy comparison between the searched network and SOTA results in a table.
3, This paper illustrates only results on VGGFace2 and RFW. Although some datasets may not have identity labels, it is necessary to show the verification/recognition performance on mainstream benchmarks, such as LFW, CFP-FP, CPLFW, AgeDB, CALFW and IJB-B/C. Experiments on two datasets are insufficient to prove the searched network achieved a good trade-off between accuracy and fairness.


**Summary Of The Paper:**

The authors ran a large-scale analysis of 29 architectures and trained them with various hyperparameters. They spent 88,493 GPU hours building this study to prove that the recognition accuracy of a network can only sometimes improve the fairness of recognition. Based on this, they conduct a neural architecture search (NAS) for fairness and accuracy. SOTA results achieved by searched networks are illustrated with Pareto pictures.

**Summary Of The Review:**

The motivation is well explained and proven. However, more results on popular mainstream benchmarks should be reported. Moreover, if the author can clarify the differences (the novelty) between the NAS they adopted and the existing SOTA NAS methods.

---

> ### Author Response · Authors · 2022-11-17
> **Response to reviewer i8qM**
>
> Thank you for referring to our experiments as extensive and our work as easy to follow for new readers. We address each of your comments below:
>
> > The proposed method seems to need more novelty. There are no new concepts or search strategies presented.
>
> To the best of our knowledge, our work is the first to extensively study the effect of hyperparameters and architectures for fairness in face recognition. Our main contribution is that architecture has a big influence on fairness, and this can be exploited to discover more fair architectures. We believe that this is impactful towards enabling and encouraging future research to study how NAS and HPO to make models fairer in a wide range of applications.
>
> We agree with the reviewer that our paper does not introduce new methods, but we respectfully point out that making the raw results from our extensive studies available and initiating NAS and HPO for fairness are novel contributions that will be very useful to the machine learning community. We would also like to mention that experimental-analysis papers have been accepted to recent ICLR, ICML, and NeurIPS conferences, and some of them have public reviews and meta-reviews available here:
>
>
> [1] Yang et al., ICLR 2020. [NAS evaluation is frustratingly hard.](https://openreview.net/forum?id=HygrdpVKvr&noteId=C5rPJ2sLzP)
>
> [2] Schmidt et al., ICML 2021. [Descending through a Crowded Valley - Benchmarking Deep Learning Optimizers.](http://proceedings.mlr.press/v139/schmidt21a.html)
>
> [3] White et al., [NeurIPS 2021. How Powerful are Performance Predictors in Neural Architecture Search?](https://openreview.net/forum?id=6RB77-6-_oI&noteId=l5H13jujQKo)
>
> [4] Ning et al., [NeurIPS 2021. Evaluating Efficient Performance Estimators of Neural Architectures.](https://openreview.net/forum?id=Esd7tGH3Spl&noteId=dl91y63zePD)
>
> [5] Mehta et-al, ICLR 2022 [NAS-Bench-Suite: NAS Evaluation is (Now) Surprisingly Easy](https://openreview.net/forum?id=0DLwqQLmqV)
>
> > It is necessary to show the verification/recognition performance on mainstream benchmarks, such as LFW, CFP-FP, CPLFW, AgeDB, CALFW and IJB-B/C. Experiments on two datasets are insufficient to prove the searched network achieved a good trade-off between accuracy and fairness.
>
> Prompted by your feedback, we have now tested our trained models (on CelebA) on the test sets suggested by you. The “SMAC” models we found via NAS for fairness outperform the competitors in all cases.
> | Architecture (trained on CelebA)        | LFW       | CFP_FF    | CFP_FP    | AgeDB     | CALFW     | CPLFW     |
> |---------------------|-----------|-----------|-----------|-----------|-----------|-----------|
> | Rexnet_200          | 71.18     | 73.62     | 54.07     | 56.31     | 61.01     | 52.22     |
> | DPN_CosFace         | 88.86     | 90.47     | 68.53     | 64.84     | 76.09     | 60.66     |
> | DPN_MagFace         | 85.88     | 89.03     | 61.30     | 60.00     | 73.50     | 55.53     |
> | DenseNet161         | 81.72     | 81.88     | 64.82     | 55.16     | 65.7      | 58.40     |
> | Ese_Vovnet39        | 73.31     | 74.42     | 63.33     | 50.00     | 59.86     | 57.93     |
> | ArcFace | 73.36     | 76.30     | 62.64     | 57.41     | 63.62     | 57.66     |
> | SMAC_000            | **94.98** | **95.60** | 74.24     | 80.23     | 84.73     | 64.22     |
> | SMAC_010            | 94.22     | 95.08     | **75.14** | **82.35** | **85.35** | **66.26** |
> | SMAC_680            | 87.45     | 90.34     | 64.22     | 61.28     | 76.16     | 56.16     |
>
> Thank you again for your thoughtful review. We made a significant effort to address each of your questions and would appreciate it if you would consider raising your score in light of our response. Please let us know if you have additional questions we can address.

---

> > ### Comment · Reviewer_i8qM · 2022-12-03
> > **Comment on authors's response**
> >
> > I would like to thank the authors for their efforts taken for the new experiments and detailed responses. In general, I am satisfied with the new results. However, my main concern has not been well addressed in the response. Similar to reviewer 7yRH, I did not find the conclusion very exciting. The manuscript lacks novelty in the NAS part. The paper mainly uses the existing approaches for a specific task. I would suggest the authors conduct more research on this topic by adding some new innovative elements to the NAS part. I maintain my original recommendation.

---

> > > ### Author Response · Authors · 2022-12-03
> > > **Further experiments and clarifications**
> > >
> > > Thank you for your feedback and your willingness to engage.  We stress that our work is not a NAS method paper;  the central point of our work is to show that architecture plays a central role in fairness despite having been widely neglected in the literature.  We feel that this contribution is practically valuable and would be of interest to the ICLR conference community as the gaps in fairness between ResNets, which are commonly used as face recognition backbones by community members, and our searched architectures are very large.  In light of our findings, any future works looking to optimize their face rec pipeline for fairness should carefully choose their architecture and hyperparameters with fairness in mind.
> > >
> > > We also want to highlight our new experiments which show that fair architectures can be combined with train-time bias mitigation strategies [1,2,3] for better fairness than either one on its own.  Notably, choosing the right architecture alone has a larger impact on reducing disparity than these three recent fair training strategies, indicating that the community should consider re-focusing their efforts towards fair face rec systems.  We include results on the Celeb-A validation and test sets with all three of our searched architectures as well as a ResNet-18 baseline below:
> > >
> > > Celeb-A validation set
> > > | Model    | Original (Acc, Disp) | [1] (Acc, Disp) | [2] (Acc, Disp) | [3] (Acc, Disp) |
> > > |----------|----------------------|------------------------|------------------------|------------------------|
> > > | Resnet-18 [4]| 80.40,  7.20         |81.25 , 3.90         | 77.15, 2.10          | 75.94, 3.70 |
> > > | SMAC_000 | **96.72**,1.57          | 96.37, 1.70           | 96.70, 1.40           | 94.81, 0.40 |
> > > | SMAC_010 | 95.77,1.89        | 95.55, 2.20           | 96.01, 1.70            |87.00, **0.23**           |
> > > | SMAC_680 | 96.68, 1.60       | 96.34, 1.62            |  96.41, 1.67           | 87.09, 5.12            |
> > >
> > >
> > > Celeb-A test set
> > > | Model    | Original (Acc, Disp) | [1] (Acc, Disp) | [2] (Acc, Disp) | [3] (Acc, Disp) |
> > > |----------|----------------------|-----------------|-----------------|-----------------|
> > > | Resnet-18 [4] | 89.70, 4.6        | 90.10, 5.60    | 90.90, 3.00     | 86.44, 2.70    |
> > > | SMAC_000 | 96.75, 2.18        | 96.55, 2.18    | 96.89, 2.28     | 94.80, **0.03**    |
> > > | SMAC_010 | 95.86, 2.27        | 95.50, 2.50    | 96.01, 2.12    | 87.63, 5.46    |
> > > | SMAC_680 | **96.78**, 1.96        | 96.20, 4.16    | 96.71, 2.09   | 87.58, 4.50    |
> > >
> > > We make the following conclusions:
> > > 1. Our strategy to search for architectures for fairness is often complementary to other bias mitigation strategies. Our approach either Pareto-dominates or is on the Pareto frontier of existing bias mitigation strategies. We highlight that using strategy [2] on our searched architectures often helps decrease disparity without significant loss of accuracy.
> > > 2. Different bias mitigation strategies are optimal for different architectures and hyperparameters.
> > > 3. Optimality of a bias mitigation strategy on the validation set does not translate directly to the test set, so we have to be careful to choose strategies which generalize.  This point, namely that bias mitigation strategies may not generalize to new test identities, is consistent with the findings of [4].
> > >
> > > We have added these findings to our local draft and will include them in our camera ready when edits on OpenReview are permitted again.
> > >
> > > We think our contributions are significant and will shift the focus of future fairness work, especially considering the scale of our improvements in disparity compared to competitors.  We have additionally made a significant effort to address each of your points, including new experiments and paper edits, and we would appreciate it if you would consider raising your score. We are very happy to engage in any further discussion.
> > >
> > > [1] [InterFace: Adjustable Angular Margin Inter-class Loss for Deep Face Recognition](https://arxiv.org/abs/2210.02018)
> > >
> > > [2] [SensitiveNets: Learning Agnostic Representations with Application to Face Images](https://arxiv.org/abs/1902.00334)
> > >
> > > [3] [On Adversarial Bias and the Robustness of Fair Machine Learning](https://arxiv.org/abs/2006.08669)
> > >
> > > [4] [Technical Challenges for Training Fair Neural Networks](https://arxiv.org/abs/2102.06764)

---

### Official Review · Reviewer_KNWY · 2022-10-27

**Confidence:** 5
**Correctness:** 2
**Technical Novelty And Significance:** 2
**Empirical Novelty And Significance:** 2
**Recommendation:** 5

**Clarity, Quality, Novelty And Reproducibility:**

The details are there for reproducibility. The concept is important but I will not rate it very high in terms of quality and novelty.

**Strength And Weaknesses:**

Strong points:
1. The usage of NAS in the context of the fairness of algorithms is novel and can provide insight into the performance of models from an architectural and hyperparameter standpoint.
2. The authors conduct extensive experimentation by utilizing 29 different model architectures and a total of 355 training runs (explained in Section 3 of the paper).
3. The authors utilize a balanced subset of the large-scale CelebA dataset for training and evaluation. They further showcase transferability results on the RFW and VGGFace2 datasets.

Weak points:
1. There is a lack of a concrete conclusion based on the experiments that have been performed in the study.
2. The authors utilize only one fairness metric, while a wide variety of metrics exist in the literature, which limits the analysis. Another popular metric, such as standard deviation across the verification accuracy of subgroups, could have been added.
3.  Several bias mitigation strategies exist in the literature. The authors have highlighted this in the Related Work section. Implementing some of those approaches and identifying the impact of hyperparameters and network architectures on face recognition performance will be an interesting addition to this study.
4. The results and analysis sections are limited.


Typos/Minor Comments:
1. The authors can provide additional information about the Rank Disparity metric stating its range and behavior.
2. In the Error vs Fairness Value curves, it can be useful to also color-code the non-optimal configurations based on the model architecture. It might provide further insight into the behavior of different models.


**Summary Of The Paper:**

In this work, the authors study the impact of different deep learning networks on the fairness of face recognition algorithms. A Neural Architecture Search is performed, and a wide variety of network architectures (with multiple sets of hyperparameters) are employed for the analysis. The Rank Disparity metric is used for fairness evaluation in the face verification task. The authors also release the code for reproducibility.


**Summary Of The Review:**

While there is always scope to train using more search strategies and databases, the authors perform extensive experimentation of the different architectures and hyperparameters for fairness in face recognition. However, a strong conclusion and key takeaways are missing from the study.

---

> ### Author Response · Authors · 2022-11-17
> **Response to reviewer KNWY - Part 1**
>
> Thank you for pointing out that our work is novel and that our experiments are extensive. Furthermore, we would like to thank you for your constructive suggestions and feedback.
>
> We respond to your questions below:
>
> > Lack of a concrete conclusion
>
> We have updated our main contributions on page 2 to be more clear and include our new results. We provide a more in depth discussion of our conclusions from the large scale analysis of architectures and hyperparameters towards the end of Section 3 on pages 4, 5 and 6. You can also find our conclusions from the joint NAS and HPO in Section 4.3. For completeness, we also state our major findings below:
>
> * We provide a new bias mitigation strategy which identifies that architectures have a profound influence on fairness, and then exploits that insight in order to design more fair architectures via Neural Architecture Search and Hyperparameter Optimization.
> * We conduct a large-scale study of 29 architectures, each trained across a variety of hyperparameters, totalling 88,493 GPU hours, showing that architectures and hyperparameters have a big impact on fairness. We then conduct the first neural architecture search for fairness, jointly with hyperparameter optimization and optimizing for accuracy — culminating in a set of architectures which Pareto-dominate all models in a large set of modern architectures.
> * Our new architectures outperform the current state of the art architecture, ArcFace, when training and testing CelebA and VGGFace2, and when training on CelebA and testing on other face recognition datasets (LFW, CFP-FP, CPLFW, AgeDB, and CALFW). Furthermore our architectures transfer well across different protected attributes Section 4.3.1.
>
> > The authors utilize only one fairness metric, while a wide variety of metrics exist in the literature, which limits the analysis. Another popular metric, such as standard deviation across the verification accuracy of subgroups, could have been added.
>
> Thank you for this suggestion!  We are very sensitive to how different fairness metrics can change the interpretation of results. In our original submission, we included many common fairness metrics for face identification (including the widely used metric from the NIST FRVT report on Demographic effects for face identification (FNIR/CRC) [1]). All our original fairness metrics we considered are listed in Table 3, and we have added new metrics in green.
>
> We chose to only focus on Rank Disparity (or as NIST FRVT calls it, FNIR disparity) in the main body of the paper because the other metrics we tested yielded similar results. This can be seen in Figures 8 through 13. We have added additional analysis on more fairness metrics in the appendix A.3 of our submission. This includes your suggested standard deviation metric and reviewer pGLd’s BPC suggestion. We believe that our results are strongly consistent across metrics with the commonly used metric form NIST FRVT as well as the other metrics we studied originally and added during the rebuttal.
>
> [1] https://nvlpubs.nist.gov/nistpubs/ir/2019/NIST.IR.8280.pdf
>
> > The results and analysis sections are limited.
>
> We have greatly increased the discussion of our results in our revision. First, we’ll highlight that, while there is no section header in Section 3, we discuss our results there from the large scale study of architectures and hyperparameters. Further in Section 4, we also provide the results from our NAS+HPO experiments in Section 4.3 where we have added content from all of our additional experiments which we have conducted based on the reviewer feedback. In addition we have added further results in the appendix section A.3 and A.4.

---

> > ### Author Response · Authors · 2022-11-17
> > **Response to reviewer KNWY - Part 2**
> >
> > > Several bias mitigation strategies exist in the literature. The authors have highlighted this in the Related Work section. Implementing some of those approaches and identifying the impact of hyperparameters and network architectures on face recognition performance will be an interesting addition to this study.
> >
> > This point is very important, and one we have included in the main portion of the paper in our updated revision. There are several face recognition-specific bias mitigation techniques that exist in the literature – many in-processing techniques which add regularizers to loss functions, adjust angle margins, introduce label noise, or learn fair representations. In the work of [1], they studied these techniques for different fair training pipelines for the ArcFace models and reported their results in Tables 6 and 7 of their work. The best bias mitigation technique from this work achieves an accuracy on Males of 93.4% and accuracy on Females of 89.1% with a performance gap of 4.3%. Our novel architecture achieves accuracies of 96-98% on both males and females which means that our bias mitigation technique outperforms those reported in [1] by a significant margin.
> >
> > Further, are currently running experiments with different bias mitigation strategies in combination with our searched architecture. We will update our response with the results when they become available.
> >
> > [1] [Cherepanova et-al Technical Challenges for Training Fair Neural Networks](https://arxiv.org/pdf/2102.06764.pdf)
> >
> > > Novelty, Difference from existing NAS methods
> >
> > To the best of our knowledge, our work is the first to extensively study the effect of hyperparameters and architectures for fairness in face recognition. Our main contribution is that architecture has a big influence on fairness, and this can be exploited to discover more fair architectures. We believe that this is impactful towards enabling and encouraging future research to study how NAS and HPO to make models fairer in a wide range of applications.
> >
> > We agree with the reviewer that our paper does not introduce new methods, but we respectfully point out that making the raw results from our extensive studies available and initiating NAS and HPO for fairness are novel contributions that will be very useful to the machine learning community. We would also like to mention that experimental-analysis papers have been accepted to recent ICLR, ICML, and NeurIPS conferences, and some of them have public reviews and meta-reviews available here:
> >
> > [1] Yang et al., ICLR 2020. [NAS evaluation is frustratingly hard.](https://openreview.net/forum?id=HygrdpVKvr&noteId=C5rPJ2sLzP)
> >
> > [2] Schmidt et al., ICML 2021. [Descending through a Crowded Valley - Benchmarking Deep Learning Optimizers.](http://proceedings.mlr.press/v139/schmidt21a.html)
> >
> > [3] White et al., [NeurIPS 2021. How Powerful are Performance Predictors in Neural Architecture Search?](https://openreview.net/forum?id=6RB77-6-_oI&noteId=l5H13jujQKo)
> >
> > [4] Ning et al., [NeurIPS 2021. Evaluating Efficient Performance Estimators of Neural Architectures.](https://openreview.net/forum?id=Esd7tGH3Spl&noteId=dl91y63zePD)
> >
> > [5] Mehta et-al, ICLR 2022 [NAS-Bench-Suite: NAS Evaluation is (Now) Surprisingly Easy](https://openreview.net/forum?id=0DLwqQLmqV)
> >
> > > The authors can provide additional information about the Rank Disparity metric stating its range and behavior.
> >
> > Rank disparity can assume values from [0,n], where n by the number of unique identities in the dataset. The value can be thought of as the absolute difference between the perceived genders of the average number of identities which are closer to a probe image than the real identity. A lower rank disparity indicates a lower fairness bias (e.g. amongst perceived genders). We adopt this metric from [1].
> >
> >  [1] Grother et-al [NIST interagency report on the evaluation of 2D still-image face recognition algorithms](https://tsapps.nist.gov/publication/get_pdf.cfm?pub_id=905968)
> >
> > > In the Error vs Fairness Value curves, it can be useful to also color-code the non-optimal configurations based on the model architecture
> >
> > From this suggestion, we have included Figure 16 which replicates the entirety of Figure 2 but with each model colored. We have also included the raw results files for analysis by other scholars who can explore further patterns in the data we collected.
> >
> > Thank you again for your thoughtful review. We made a significant effort to address each of your questions and would appreciate it if you would consider raising your score in light of our response. Please let us know if you have additional questions we can address.

---

> > > ### Author Response · Authors · 2022-12-03
> > > **Followup experiments and further discussion**
> > >
> > > Thanks again for your feedback! We wanted to provide you with the results of new experiments that address your remaining point, which show that our novel fair architecture alone has a larger impact on fairness than recent fair training methods.
> > >
> > > > Several bias mitigation strategies exist in the literature. The authors have highlighted this in the Related Work section. Implementing some of those approaches and identifying the impact of hyperparameters and network architectures on face recognition performance will be an interesting addition to this study.
> > >
> > > We agree that including existing bias mitigation strategies is a valuable addition to our work, and prompted by your suggestion, we have now implemented three commonly used bias mitigation strategies [1,2,3] from the literature in combination with our searched models.  Interestingly, we find that some bias mitigation strategies are orthogonal to our architectural improvements and can be used in combination with our searched architectures for further boosts in fairness at little accuracy cost.  But more importantly, choosing the right architecture can itself yield more improvement to disparity than these three fair training strategies, indicating that the community should consider re-focusing their efforts towards fair face rec systems.  We include results on the Celeb-A validation and test sets with all three of our searched architectures as well as a ResNet-18 baseline below:
> > >
> > > Celeb-A validation set
> > > | Model    | Original (Acc, Disp) | [1] (Acc, Disp) | [2] (Acc, Disp) | [3] (Acc, Disp) |
> > > |----------|----------------------|------------------------|------------------------|------------------------|
> > > | Resnet-18 [4]| 80.40,  7.20         |81.25 , 3.90         | 77.15, 2.10          | 75.94, 3.70 |
> > > | SMAC_000 | **96.72**,1.57          | 96.37, 1.70           | 96.70, 1.40           | 94.81, 0.40 |
> > > | SMAC_010 | 95.77,1.89        | 95.55, 2.20           | 96.01, 1.70            |87.00, **0.23**           |
> > > | SMAC_680 | 96.68, 1.60       | 96.34, 1.62            |  96.41, 1.67           | 87.09, 5.12            |
> > >
> > >
> > > Celeb-A test set
> > > | Model    | Original (Acc, Disp) | [1] (Acc, Disp) | [2] (Acc, Disp) | [3] (Acc, Disp) |
> > > |----------|----------------------|-----------------|-----------------|-----------------|
> > > | Resnet-18 [4] | 89.70, 4.6        | 90.10, 5.60    | 90.90, 3.00     | 86.44, 2.70    |
> > > | SMAC_000 | 96.75, 2.18        | 96.55, 2.18    | 96.89, 2.28     | 94.80, **0.03**    |
> > > | SMAC_010 | 95.86, 2.27        | 95.50, 2.50    | 96.01, 2.12    | 87.63, 5.46    |
> > > | SMAC_680 | **96.78**, 1.96        | 96.20, 4.16    | 96.71, 2.09   | 87.58, 4.50    |
> > >
> > > We make the following conclusions:
> > > 1. Our strategy to search for architectures for fairness is often complementary to other bias mitigation strategies. Our approach either Pareto-dominates or is on the Pareto frontier of existing bias mitigation strategies. We highlight that using strategy [2] on our searched architectures often helps decrease disparity without significant loss of accuracy.
> > > 2. Different bias mitigation strategies are optimal for different architectures and hyperparameters.
> > > 3. Optimality of a bias mitigation strategy on the validation set does not translate directly to the test set, so we have to be careful to choose strategies which generalize.  This point, namely that bias mitigation strategies may not generalize to new test identities, is consistent with the findings of [4].
> > >
> > >
> > > We have added these findings to our local draft and will include them in our camera ready when edits on OpenReview are permitted again.  Thank you again for your feedback!  We have made a significant effort to address each of your points, including new experiments and paper edits, and we would appreciate it if you would consider raising your score. We are very happy to engage in any further discussion.
> > >
> > >
> > > [1] [InterFace: Adjustable Angular Margin Inter-class Loss for Deep Face Recognition](https://arxiv.org/abs/2210.02018)
> > >
> > > [2] [SensitiveNets: Learning Agnostic Representations with Application to Face Images](https://arxiv.org/abs/1902.00334)
> > >
> > > [3] [On Adversarial Bias and the Robustness of Fair Machine Learning](https://arxiv.org/abs/2006.08669)
> > >
> > > [4] [Technical Challenges for Training Fair Neural Networks](https://arxiv.org/abs/2102.06764)

---

> ### Author Response · Authors · 2022-12-08
> **Follow up for additional engagement**
>
> As we come to the end of Discussion Stage 2, we are happy to clarify any additional points.  We have conducted extensive additional experiments, including ones which demonstrate the superiority of our approach to existing bias mitigation strategies from the literature, namely simply swapping out the architecture of a face rec system improves fairness metrics more so than specially designed fair training routines.  This result will likely impact the direction of future fairness research as the inductive biases of our models are evidently more influential than fairness interventions during training.  Our additional experiments also address each of the points raised by reviewers where appropriate as do the edits we made to our updated draft.  Reviewer pGLd has raised their score to an 8 in light of our response, and we have also responded to each other reviewer equally comprehensively.  Do you have any additional questions we can address?

---

### Author Response · Authors · 2022-11-17
**General Response**

We thank the reviewers for their thoughtful and constructive feedback on improving our paper. We here provide a general response, addressed to all reviewers and ACs, as well as individual replies to address specific reviewer concerns as separate posts.

In our responses, we have conducted extensive experiments and benchmarking of our new architecture and found its superior performance extends to new datasets and new metrics. First, we see that our architecture is state of the art on CelebA for face identification. Additionally, we show that our architecture outperforms the current state of the art ArcFace architecture on VGGFace2. Finally, its performance is superior to all other architectures we trained on CelebA when transferred to different face recognition datasets like LFW, CFP-FP, CPLFW, AgeDB, and CALFW.

We highlight the main changes to the manuscript below:
* We have improved the presentation of our contributions and results throughout
* We have now added results on more face recognition benchmarks in Section A.4
* We add analysis on several metrics to section A.3 of the appendix
* We train our models on VGGFace2 and provide the results in section A.5 of the appendix.
(Major changes in the manuscript are highlighted in green.)

As pointed out by several reviewers our work lays a strong foundation for the study of the impact of architectures and hyperparameters for fairness in face recognition. The main contribution of our work is that we can exploit the insight that architecture has a big influence on fairness in order to make more fair architectures. The impact of this technique is powerful and will be broadly useful to a range of scholars in order to make more fair models on a range of topics.

Though a plethora of research has proposed several pre-processing, post-processing techniques, and fair training pipelines, to the best of our knowledge none of them study if the architectures and hyperparameters chosen themself impact the fairness of architectures. Given that architectures are traditionally handcrafted, we seek to answer the question if using Neural Architecture Search (NAS) and Hyperparameter optimization (HPO) to automatically discover these architectures can help overcome their design bias. We support our findings on the importance of architectures and hyperparameters with extensive experiments amounting to 88k GPU hours on different image classification models ranging from Inception networks to Vision Transformers. Further motivated by our findings we initiate the first study of using joint NAS and HPO for fairness in face recognition systems. Finally, our work aims at empowering the domain expert (in fairness for face recognition) with the appropriate evidence and tools to discover inherently fair architectures and hyperparameters.

With this we hope we have addressed a majority of the concerns, if there are concerns still pending, please feel free to reach out to us.

---

### Author Response · Authors · 2022-12-08
**Check-in regarding outstanding concerns or questions**

We again thank the reviewers and AC for their time and feedback.  Since the end of Discussion Stage 2 is drawing near, we would like to take any last opportunities to clarify additional points. Any engagement with our author responses is greatly appreciated.  We have conducted extensive additional experiments, including ones which demonstrate the superiority of our approach to existing bias mitigation strategies from the literature, namely simply swapping out the architecture of a face rec system improves fairness metrics more so than specially designed fair training routines.  This result will likely impact the direction of future fairness research as the inductive biases of our models are evidently more impactful than fairness interventions during training.  Our additional experiments also address each of the points raised by reviewers where appropriate as do the edits we made to our updated draft. Reviewer pGLd has raised their score to an 8 in light of our response, and we have also responded to each other reviewer equally comprehensively.  Please let us know, by responding to our individual responses, if there are any additional questions you have.

---

### Decision · Program_Chairs · 2023-01-20

**Decision:**

Reject

**Justification For Why Not Higher Score:**

Novelty is the main concern. This paper focuses on empirical study, it offers good guidance to pickup architecture for face recognition under accuracy-fairness trade-off. However, it does not exhibit new technologies or insights. In its current form, NeurIPS benchmark track or TMLR can be a better place to go.

**Justification For Why Not Lower Score:**

N/A

**Metareview: Summary, Strengths And Weaknesses:**

This paper considers the impact of network architecture and hyperparameter under accuracy-fairness metric for face recognition. The paper adopts existing approaches to evaluate different models under accuracy-fairness metric. Extensive experiments have been performed with empirical suggestions are made.

Strength
- The presentation is easy to follow.
- Experiments are extensive.

Weakness
- Novelty (main issue). It is obvious that network architectures and hyperparameters can have great impact on predicting performance, and many existing works have study similar problems (see [1] for an example). Compared with existing benchmark / understanding approaches, the new things introduced is "fairness" metric. However, no new metric is proposed. Besides, it is not clear what "special" insights can be generated for "fairness".
- Insufficient comparison. Existing NAS methods should be compared. There are popular types of NAS methods that can handle the discrete "Rank disparity" metric. For example, the authors may have a check over [2].


[1]. Graph Structure of Neural Networks. ICML 2020.
[2] Single Path One-Shot Neural Architecture Search with Uniform Sampling. ECCV 2020.